# Effect of Sustainably Sourced Protein Consumption on Nutrient Intake and Gut Health in Older Adults: A Systematic Review

**DOI:** 10.3390/nu16091398

**Published:** 2024-05-06

**Authors:** Debra Jones, Carlos Celis-Morales, Stuart R. Gray, Douglas J. Morrison, Susan E. Ozanne, Mahek Jain, Lewis R. Mattin, Sorrel Burden

**Affiliations:** 1School of Health Sciences, University of Manchester, Oxford Road, Manchester M13 9PL, UK; sorrel.burden@manchester.ac.uk; 2School of Cardiovascular and Metabolic Health, University of Glasgow, Glasgow G12 8TA, UK; carlos.celis@glasgow.ac.uk (C.C.-M.); stuart.gray@glasgow.ac.uk (S.R.G.); mahek.jain@glasgow.ac.uk (M.J.); 3Scottish Universities Environmental Research Centre (SUERC), University of Glasgow, Glasgow G75 0QF, UK; douglas.morrison@glasgow.ac.uk; 4Metabolic Research Laboratories and MRC Metabolic Diseases Unit, Institute of Metabolic Science, University of Cambridge, Addenbrookes Hospital, Cambridge CB2 0QQ, UK; seo10@cam.ac.uk; 5School of Life Sciences, University of Westminster, London W1W 6UW, UK; l.mattin@westminster.ac.uk; 6Salford Care Organisation, Northern Care Alliance NHS Trust, Stott Lane, Salford M6 8HD, UK

**Keywords:** healthy ageing, gut microbiota, gut health, sustainable diet, sustainably sourced protein, plant based diet

## Abstract

Diet is integral to the healthy ageing process and certain diets can mitigate prolonged and deleterious inflammation. This review aims to assess the impact of diets high in sustainably sourced proteins on nutrient intake, gut, and age-related health in older adults. A systematic search of the literature was conducted on 5 September 2023 across multiple databases and sources. Studies assessing sustainably sourced protein consumption in community dwelling older adults (≥65 years) were included. Risk of bias (RoB) was assessed using ‘RoB 2.0′ and ‘ROBINS-E’. Narrative synthesis was performed due to heterogeneity of studies. Twelve studies involving 12,166 older adults were included. Nine studies (n = 10,391) assessed habitual dietary intake and had some RoB concerns, whilst three studies (n = 1812), two with low and one with high RoB, conducted plant-based dietary interventions. Increased adherence to sustainably sourced diets was associated with improved gut microbial factors (n = 4640), healthier food group intake (n = 2142), and increased fibre and vegetable protein intake (n = 1078). Sustainably sourced diets positively impacted on gut microbiota and healthier intake of food groups, although effects on inflammatory outcomes and health status were inconclusive. Future research should focus on dietary interventions combining sustainable proteins and fibre to evaluate gut barrier function and consider inflammatory and body composition outcomes in older adults.

## 1. Introduction

Ageing is characterised by deterioration of physical and mental capacity and function over time [1]. In addition, as people age, bhthey become more susceptible to a wide range of diseases, which can lead to a decline in health and wellbeing [2]. In 2020 1 billion people worldwide were aged 60 or over. It is expected that this number will have increased to 1.4 billion by 2030, and to have more than doubled to 2.1 billion by 2050 [2]. In the UK the number of people aged 65 years and over has increased from 9.2 million in 2011 to over 11 million in 2021 [3]. Of these, 12.8% reported being in bad or very bad health and the likelihood of reporting good health decreases with age in this population [3]. Declined health in older age may include, but is not limited to, reduced mobility, functional decline, under-nutrition, dehydration, loss of independence, loss of bone health and incontinence. This reduced health can be referred to as disabilities in older age. A recent report proposed changes to the focus of health care to concentrate on quality of life rather than number of years lived in older people [4]. To maximise quality of life in older people it is key to reduce the accumulation of comorbidities and maintain functionality whilst decreasing years of life lived with disabilities. Increasing disability free years will therefore need to focus on prevention and management of multimorbidity with maintenance of functionality.

Evidence suggests that low-grade inflammation, induced during the ageing process, plays an important role in the onset of a wide range of age-related diseases including sarcopenia, dementia, type 2 diabetes, atherosclerosis, vascular disease, obesity, arthritis, cancer, and osteoporosis [5,6]. In addition to low grade inflammation, disruption of the gut microbiota (known as dysbiosis) occurs as part of the ageing process and is accelerated by multiple factors including changes in gut physiology, and dietary intake [7] and changes in the hormonal milieu, which can include reduced testosterone in males, lower estrogen in females after menopause and gut-hormonal related changes in ghrelin [8,9]. Dysbiosis reduces the capacity of the microbiota to carry out metabolic processes such as short chain fatty acid production, which is a contributing factor to increased inflammation in older adults (inflamm-ageing) [7,10,11,12,13]. It has also been suggested that these changes and deterioration of the gut microbiome in older age, may lead to changes in protein metabolism, absorption and bioavailability [14], resulting in reduced muscle protein synthesis [7], frailty [15] and sarcopenia [16]. 

Diet can play an important role in healthy ageing as certain nutrients can modulate acute and chronic inflammation [17]. Consumption of a healthy diet, including wholegrains, vegetables, fruit and fish, are associated with lower circulating inflammatory markers [18,19,20,21,22,23]. Consumption of high-quality protein and dietary fibre can also help to improve metabolic health, body weight, gut motility and inflammation [24,25] as well as promoting growth of gut bacteria and preserving a healthy gut microbiota and gut barrier function [7,14,16]. Maintaining sufficient dietary protein and fibre intake are important in the context of maintaining muscle strength and physical function as individuals age [26,27]. There is also evidence that older adults often eat insufficient protein and fibre and consequently have a lower diet quality than required, contributing to impaired physical function [26]. 

Animal proteins, including egg, milk, poultry, fish and meat, are considered high-quality proteins as they contain all nine indispensable amino acids (IAAs) in high quantities and at high bioavailability [28]. They have therefore often been considered to be superior to plant-based proteins for promoting optimal muscle protein synthesis [29]. However, more recent work suggests that this may not be the case if sufficient protein is consumed from a variety of plant-based sources [30,31,32,33]. Plants tend to have lower protein by mass, lower bioavailability, contain antinutritive factors and can be deficient in some IAAs if reliant on a single plant source. Consequently, a plant-based diet needs to be complemented with other plant, vegetable, dairy and meat based proteins to balance bioavailability of IAAs against requirements [34]. An important additional factor is that plant-based protein intake is associated with increased dietary fibre intake whereas animal-based protein is associated with reduced intake of fibre [35], has detrimental impacts on the environment [36], and is more expensive [37]. Hence, there is a need to support older people in maintaining or increasing their protein and fibre intake using sustainably sourced foods.

A sustainable diet should take into consideration all dimensions of sustainability. The WHO and FAO have agreed on a two dimensional term of ‘sustainable healthy diets’ and stated that this refers to ‘dietary patterns that promote all dimensions of individuals health and wellbeing; have low environmental pressure and impact; are accessible, affordable, safe and equitable; and are culturally acceptable’ [38]. The EAT Lancet commission firstly defines a universal healthy diet as: ‘largely consists of vegetables, fruits, whole grains, legumes, nuts, and unsaturated oils, includes a low to moderate amount of seafood and poultry, and includes no or a low quantity of red meat, processed meat, added sugar, refined grains, and starchy vegetables’ and secondly state that ‘sustainable food production stays within safe planetary boundaries for six environmental processes that together regulate the state of the Earth system, and include climate change, land-system change, freshwater use, biodiversity loss, and interference with the global nitrogen and phosphorus cycles’ [39]. For the purposes of this review, the definition of sustainably sourced protein refers to sustainable or plant-based diets due to the positive environmental effects of replacing animal sourced foods with plant-based foods [39]. In addition, we also consider vegan, vegetarian and Mediterranean diets, or any diet of a similar nature, to be sustainable diets due to their lower content of animal-based foods and association with reductions in greenhouse gases, land use and water use [39]. However, it is important to highlight the contrast of scientific evidence and consumer perceptions surrounding sustainable diets. A recent survey, conducted in 420 German city-dwellers (in their 20s and 30s), concluded that there was a science-belief gap with participants associating regional, seasonal and organic foods with sustainable diets rather than plant-based or meat-free foods [40]. Therefore, it will be essential to improve consumer knowledge to achieve future goals for sustainable diets and food systems.

Previous systematic reviews have focused on the effect of consuming different types of protein on muscle health [41,42,43,44,45,46], gut health [47,48,49] or disease risk or status [50,51]. Further investigation is warranted as there is a paucity of data linking gut health outcomes with muscle health or age-related health outcomes. Furthermore, most reviews have focused on certain types of proteins including protein supplements [45], soy [42], dairy [46], meat [49], beef [48], or specific diets such as vegan and/or vegetarian [47]. There is a lack of consideration given to sustainable proteins or sustainable diets and their associated fibre intake. In addition, several of these reviews were not specific to older adults [42,43,47,48,49,50] and one excluded participants over the age of 40 years [41]. 

This systematic review addresses gaps in the current evidence base by specifically focusing on the effect of increased plant-based protein consumption in older age and widening outcomes by including nutrient intake (particularly fibre), gut, and health outcomes. 

The aim of this systematic review is to evaluate the literature on the effect of consumption of sustainably sourced proteins compared to non-sustainable or meat-based proteins on nutrient intake, gut health (defined by gut microbiota or inflammatory markers) and age-related changes in food intake and disease status in healthy, community-dwelling individuals aged 65 years and over. This review followed the PRISMA 2020 guidance for reporting systematic reviews and the PRISMA 2020 checklist for this review can be found in Appendix A.

## 2. Methods

### 2.1. Eligibility Criteria

#### 2.1.1. Population

We included randomised controlled trials, prospective and retrospective cohort studies, observational studies and cross-sectional studies that examined the older adult human population or healthy older adult humans with an inclusion criterion of 65 years and older. We also included studies on people who were underweight, overweight or obese but otherwise excluded studies of populations restricted to specific diseases, conditions, or metabolic disorders. We included studies addressing adults in general if data provided for older adults (≥65 years) were reported separately.

#### 2.1.2. Interventions

Interventions or exposures included were any dietary adherence or intervention that could be considered plant-based or sustainable. The criteria used to define the diet had to have evidence of being sustainably sourced or plant-based or demonstrate that there is a dietary change towards being more sustainable or plant-based. These could include the following: sustainable diets in combination with any other nutrients including fibre, fat, sugar, and carbohydrates; plant-based protein diets, including legumes or pulses, soy, wheat, vegetables, potatoes, beans, peas, quinoa, amaranth, buckwheat, rapeseed oil, edible insects, algae (seaweed), microalgae (spirulina), aquatic plants (duckweed) [52], whole diet interventions where the diet was predominantly plant sources (vegetarian/vegan); whole diet interventions where protein is predominantly from sustainable sources or where the diet has a decreased consumption of red meat with concomitant increase in sustainably-sourced proteins or a diet that is predominantly sustainably-sourced but may still include other types of protein (Mediterranean, vegetarian, plant-based). Additionally, exposures to dietary educational resources, dietary public health policies, or a populations habitual diet were included if they were related to sustainably sourced diets/proteins [52]. Studies including other lifestyle components, such as physical activity, were only included if the effects of the diet component were reported independently. Examples of interventions that we did not include were those with supplements or supplementation of concentrated forms of plant protein (plant protein powder). The minimum intervention or exposure period was 2 weeks, which has been shown to be sufficient to improve the metabolic status by a decline in potentially pro-inflammatory components (Collinsella), and increase in leanness-related taxa and reversal of gut microbiota dysbiosis in older obese women [53]. Additionally, a recent systematic review presented evidence that short term dietary interventions, ranging from 2 days to 12 weeks, have significant effects on gut microbiota so it is important to capture these data as well as data from any long term interventions [54]. 

#### 2.1.3. Comparators

Comparators included control or non-plant-based diets, or level of adherence to a sustainable or plant-based diet. 

#### 2.1.4. Outcomes

To be included, a study had to record a measurement of gut microbiome/inflammation status, or intake of individual nutrients or intake of individual food groups. We also recorded health and quality of life outcomes to assess the associated change in age-related health, although these outcomes were not required for study inclusion. Our primary outcome was the effect of a sustainably sourced, plant-based diet on nutrient intake and gut health (measured by gut microbiota or inflammatory markers). Outcomes were extracted as reported, with the exception of quality of life, which was collected only if assessed with generic (not disease specific), validated tools. Data were extracted for outcomes in all data forms (e.g., dichotomous, continuous) as reported in the included studies.

#### 2.1.5. Settings

Only studies with community dwelling older people were included.

#### 2.1.6. Years, Language, and Publication Status

Articles published anytime up to present day were included, reported in any language and articles that described original research published in peer-reviewed scientific journals. A summarised overview of inclusion and exclusion criteria can be viewed in Appendix A. 

### 2.2. Information Sources

On 5 September 2023 searches were conducted on six health and social care databases with results limited to humans only. The names and date coverage of all databases searched are given in Appendix A. Search strategies were run simultaneously for four databases in Ovid and the results de-duplicated using the Ovid de-duplication tool (web application, Ovid technologies, Inc., New York, NY, USA, 2024). To identify any ongoing trials, we also searched clinical trial registries; ISRCTN registry and ClinicalTrials.gov. On the same date we also ran a general internet search using google (http://www.google.com), browser incognito mode, and the search string: (plant based OR sustainable diet) AND (gut health OR microbiota OR microbiome) AND (older age OR elderly OR older OR ageing OR aging OR over 50), the first 20 results were selected and screened. Finally, on the 11 of October 2023, we scanned and manually screened the reference and citation lists of all included studies and any relevant reviews that had been identified through the search. 

### 2.3. Search Strategy

Literature search strategies were developed using medical subject headings (MeSH) and text words related to sustainable protein, gut microbiota, age-related diseases and older people. The full search strategies for all databases and registries can be found in Appendix A.

### 2.4. Selection Process

Two reviewers (D.J. and S.B.) independently screened the titles and abstracts against the inclusion criteria. The full text was obtained for any studies that met the inclusion criteria or where there was any uncertainty. Review authors screened the full text reports and decided whether they met the inclusion criteria. We did not need to contact any study authors to clarify eligibility. Disagreements between the reviewers were resolved through discussion and arbitration with a third author (D.J.M.) if necessary. The reasons for excluding trials were noted and ordered.

### 2.5. Data Collection Process

Literature search results were uploaded to Endnote, v20.3 for organisation and then Rayyan software (web application) for screening. Any remaining duplicates were identified and removed automatically by both Endnote and Rayyan software. Data were extracted from included studies using a standardised data extraction form, developed a priori. Data extraction was carried out by one reviewer (D.J.) and then verified by another (S.B.) to reduce bias and errors. Any disagreements were resolved through discussion and a third author consulted for arbitration when necessary. 

### 2.6. Data Outcomes

#### 2.6.1. Outcomes

The main outcomes of interest were change in nutrient and food group intake and change in gut microbiome. To measure these changes, eligible outcomes were broadly categorised as follows:Dietary adherence or intervention: Score and adherence to a particular dietNutrient intake: Energy, fat, carbohydrates, total protein, vegetable protein, animal protein and fibre.Food Groups: Meats, fish and poultry; fruit and vegetables; legumes; and breads, grains and cereals.Gut microbiome: Measurement of microbiota and inflammation status including but not limited to inflammatory markers, α-diversity, taxonomies, phenolic profiles, microbial levels or short chain fatty acids

There were no restrictions placed on the number of timepoints at which the outcomes were measured nor were there restrictions on length of follow up, as current evidence demonstrates that there are benefits to short-term dietary interventions on the gut microbiome and the impact of long-term dietary interventions is still unclear [54].

In addition, we recorded associated change in age-related health by extracting the following data variables:Health: Body mass index (BMI), self-rated health, non-communicable diseases, cholesterol, muscle mass and grip strength.

#### 2.6.2. Other Variables

Other data variables were:Author, year, and countryStudy design, sample size, mean age, gender, dietary exposure, duration of follow up.Dietary measurement tool, dietary pattern assessment, adherence to diet.

### 2.7. Study Risk of Bias Assessment

Risk of bias was assessed in the included studies using the revised Cochrane ‘risk of bias’ tool; known as Risk of Bias 2.0 (RoB 2.0) [55] for randomised controlled trials and the Risk of Bias in Non-Randomised Studies of Exposures (ROBINS-E) [56] for cross-sectional and cohort studies. Two review authors (D.J. and S.B.) independently applied the appropriate tool to each included study and recorded supporting information and justifications for judgements of RoB. Within each domain, a series of questions (‘signalling questions’) were answered to elicit information about features of the trial relevant to RoB. A proposed judgement about the RoB arising from each domain and for each study was generated by an algorithm, based on answers to the signalling questions. Judgement was either ‘Low’ or ‘High’ risk of bias, or ‘Some concerns’. Any discrepancies in judgements of risk of bias or justifications for judgements were resolved by discussion to reach consensus between the two review authors, with a third review author (D.M.) acting as an arbiter if necessary. 

### 2.8. Effect Measures

It was not possible to pool the data statistically using meta-analysis due to the low number of studies and heterogeneity in reporting, and so a narrative synthesis of the results was conducted. As such we did not conduct assessments for reporting bias or certainty of evidence.

## 3. Results

### 3.1. Study Selection

We found a total of 5084 records from searching databases and trial registers. After removing duplicates, 4485 records were screened. From these 36 full-text documents were reviewed and seven studies were included. Additionally, a further 13 records were identified from the Google search, and from scanning the reference lists of initially included studies. After checking for duplicates we reviewed the 13 additional reports and included five, resulting in a total of 12 included reports [57,58,59,60,61,62,63,64,65,66,67,68], from 11 different studies. The PRISMA flow diagram can be viewed in Figure 1. Of the full-text documents reviewed (n = 49), 37 were excluded and reasons were listed on the PRISMA flow diagram in Figure 1. We did not find any articles in different languages and so had no need for translation. We also did not need to contact any authors for further information/clarification.

### 3.2. Study Characteristics

#### 3.2.1. Population and Settings

Across the 12 papers there were 12,166 older people (≥65 years), of these 3267 were female, 4453 were male, and 4446 are of unknown sex as three studies [61,65,67], which included other adult age ranges, did not report sex split for the ≥65 years subgroup. Eight studies (n = 7661) reported mean ages (SD) ranging from 68.7 (6.4) to 84.3 (4.1) years [57,58,59,60,62,63,64,66], three studies included other adult age ranges in their sample, not just older people, and as with sex did not provide a mean age for the ≥65 years subgroups [61,65,67], and one study split mean age (SD) by sex; female: 77.7 (7.6) years, n = 30 and male: 85.3 (8.4) years, n = 19 [68]. It is also important to note that one study collected gut microbiota outcomes from older people with a mean age of 69.2, but the dietary intake data from these participants had been collected 20 years earlier [64]. Included studies were conducted across nine countries; two studies used different data from the same trial, which was gathered from five different European countries (France, The Netherlands, UK, Poland and Italy) [58,60]; four studies were conducted in the United States of America [59,62,64,66]; three in Spain [61,63,65]; and one each in Greece [67], Taiwan [68] and France [57]. Further details for the study and population characteristics can be found in Table 1.

#### 3.2.2. Study Design and Intervention

Of the 12 papers included in the review, five were cross-sectional studies [57,59,61,65,66], where usual dietary intake was measured at one time point and then either measured for daily animal and plant protein intake [59]; or scored for adherence to a particular diet(s), including Mediterranean [57,61], prudent [57,66], traditional/Western [57,66], and complex [57]; or scored for adherence to particular dietary indices [65] (dietary inflammatory index, empirical dietary inflammatory index, healthy eating index, alternative healthy eating index, Mediterranean adapted diet quality index-international, modified Mediterranean diet score, and relative Mediterranean diet score). A further five studies were cohort studies [62,63,64,67,68] and four of these measured usual dietary intake at two or more time points [62,63,64,67] and then scored for adherence to a particular diet or dietary index, including healthy plant-based index [62,63], unhealthy plant-based index [63], healthy eating index 2010 [64], alternative healthy eating index 2010 [64], alternate Mediterranean diet [64], dietary approaches to stop hypertension trial (DASH) diet [64], and the Mediterranean diet [67]. The one remaining cohort study used a plant-based smoothies and snacks intervention, which provided participants with 8 weekly servings of smoothies and snacks over a 4-month period and compliance to the intervention was measured. Two out of the 12 studies were randomised, multicentre, single-blind, controlled trials [58,60] and both used data from the same over-arching trial (NU-AGE randomized trial). The NU-AGE trial intervention was a Mediterranean-like diet (NU_AGE diet) with counselling and dietary advice compared to control, followed over 1 year with outcomes measured and baseline and 1 year. 

To measure dietary intake; eight studies used food frequency questionnaires [57,59,61,62,64,65,66,67], two studies (using data from same trial) used self-reported 7-day food diaries [58,60], one used a computerised face-to face diet history [63], and one measured the compliance to consumption of plant-based smoothies and snacks [68]. Further details for study design and intervention characteristics can be found in Table 1 and Table 2.

### 3.3. Risk of Bias in Studies

The RoB 2.0 tool was used to assess RoB in included randomised control trials and the ROBINS-E tool to assess RoB in cross-sectional and cohort studies. A summary of these assessments is provided in Table 3. 

In terms of overall RoB, there were no concerns about the two papers assessed by the RoB 2.0 tool [58,60], which both came from the same study and in-depth details were provided for all aspects of the study including appropriate randomisation processes. There were concerns about risk of bias for all studies assessed by the ROBINS-E tool [57,59,61,62], with one assessed at high risk of bias [68]. A summary text is provided below for each of the seven domains with the ROBINS-E assessment.

#### 3.3.1. Bias Due to Confounding

Some concerns were raised in this domain for five of the assessed studies [59,61,62,66,68], where there was a lack of control for all of the confounding factors that were thought to be important, however, it was noted for all five studies that the uncontrolled confounding was probably not substantial.

#### 3.3.2. Bias Arising from Measurement of the Exposure

This was the most common criteria for which risk of bias was apparent with concerns raised for all 10 studies. This was mainly due to the likelihood of error when measuring dietary exposure, which required self-recall or self-report from the participants. One study was considered high risk as the exposure was the addition of plant-based snacks and smoothies to regular diet. As no account was taken for the whole diet, we considered this exposure to not well-characterise the exposure metric of interest for this review [68].

#### 3.3.3. Bias in Selection of Participants into Study

Risk of bias was considered low for all 10 studies as participants were selected appropriately and within the specified exposure window. 

#### 3.3.4. Bias Due to Post-Exposure Interventions/Missing Data/Measurement of the Outcome/Selection of the Reported Result

Bias was considered to be low in all four of these domains for all 10 studies. Upon reviewing these studies, we felt that there were no post-exposure interventions, data were complete for all studies, outcomes were measured accurately and appropriately and all studies provided an analysis plan and reported accordingly. 

### 3.4. Results of Individual Studies

#### 3.4.1. Adherence to Diet

All studies recorded adherence or compliance to a particular type of plant-based, high protein sustainable diet. Table 2 displays the detailed results. Nine studies (n = 10,391) measured participants usual dietary intake and then assessed for compliance to sustainable-type diets [57,59,61,62,63,64,65,66,67], and three studies (n = 1812) provided a dietary intervention to participants, asking them to either follow the NU-AGE (Mediterranean style) diet [58,60] or consume plant-based smoothies and snacks [68]. Two out of these three also provided dietary counselling and free healthy foods as part of the intervention [58,60]. 

Of the nine studies that measured usual diet and recorded compliance to sustainable-type diets, five reported scores associated with the Mediterranean diet or index with three studies reporting that 21.5% [67], 26.8% [64] and 35.0% [57] of participants had high compliance or greater adherence to the diet; one study did not report compliance for the >65 years subset [61]; and one study reported mean scores for 3 different Mediterranean diet indices but unlike other studies did not divide the overall score into higher or lower levels of compliance [65]. A further three studies, measuring usual diet and recording compliance, reported either compliance with a healthy plant-based diet with high compliance from 19.8 to 36% [62,63] or the level of compliance with a high protein diet (high compliance: 25.1%) [59]. The final study measuring usual diet recorded compliance to the prudent diet, a diet high in fruits, vegetables, nuts, fish and poultry and reported that 25% of participants had a high compliance to this diet.

The two studies that used the NU-AGE dietary intervention reported that at 1 year follow up the mean score for compliance with the diet increased more in the intervention group (from 82.6 to 105.7) compared to the control group (from 82.6 to 84.6) [58], and the intervention group had significantly higher intake fibre, vitamins (C, B6, B9, thiamine) and minerals (Cu, K, Fe, Mn, Mg) compared to control, which had significant increases in fat intake compared to intervention group [60].

The one study that used a plant-based smoothies and snacks intervention, reported that 88% of female participants and 92.5% of male participants complied with consumption of five servings of smoothies and three servings of snacks per week, for a 4-month period [68]. 

#### 3.4.2. Microbiota/Inflammatory Outcomes

Nine out of the 12 studies measured gut microbial factors (n = 4754) [57,59,60,61,62,64,65,66,68] with seven of these measuring microbiome diversity [59,60,62,64,65,66,68], two measuring short chain fatty acids (SCFAs) [65,68], one measured 3-hydroxy fatty acids (3-OHFA) [57] and one phenolic acid profiles [61]. However, none of the studies reported on inflammation or inflammatory markers. Six of these studies (n = 4640) demonstrated that increased adherence to sustainably sourced diets was associated with improvements in gut microbiome diversity [59,60,62,64,66] or 3-OHFA [57] with further details provided below.

Of the seven studies (n = 4041) that measured microbiome diversity, five studies (n = 3942) demonstrated diversity improvements associated with sustainably sourced diets [59,60,62,64,66]. Four reported alpha and beta diversity with two reporting increased diversity (Shannon and Simpson index) with increased adherence to sustainable-type diets [59,64], one reporting significant associations of beta diversity to prudent dietary intake [66], and one (n = 59) reporting a decrease in species richness (Chao1 and Abundance-based coverage estimators indices) at months 2 and 4 of the intervention relative to baseline [68]. Two of the seven that measured microbiome diversity, reported specific taxa and stated that adherence to sustainable-type diets was associated with higher diversity [60,62]. One study reported specific microbial levels but as this report did not separate these results between age groups, it was not possible to ascertain the impact of the diet in the ≥65 years sub-group [65].

Of the two that measured SCFAs one (n = 40) identified that SCFAs were lower in the ≥ 65 years subgroup compared to the 50–65 years subgroup but did not compare this with dietary adherence [65], the other study (n = 59) found no significant changes in SCFAs after 2 and 4 months of plant-based smoothies and snacks intervention [68]. 

The one study that measured 3-OHFAs (n = 698) reported that higher adherence to a plant-based diet was associated with lower 3-OHFAs, with mean (SD) levels of 276.7 (110.4) pmol/mL being associated with low adherence to the Mediterranean diet and 263.7 (99.7) pmol/mL being associated with high adherence [57]. 

The one study that measured phenolic profiles reported that older subjects (>65 years), compared to younger (50–65 years) had higher values for the fecal content of phenylacetic, 4-hydroxyphenylacetic, and phthalic acids, but this study did not compare the outcomes of the >65 years subgroup with dietary adherence [61]. See Table 4 for further details.

#### 3.4.3. Food Group Intake

Three out of the 12 studies (n = 2142) measured intake of different food groups, two studies measured intake according to adherence to sustainable-type diets [57,62] and one measured the change in consumption of food groups from baseline to follow up at 1 year of the NU-AGE dietary intervention, in both intervention and control groups [58].

Of the two studies that measured intake according to adherence to diet, both reported that higher adherence to a sustainable-type diet was associated with increased consumption of fruit, vegetables, legumes, fish and wholegrains, and decreased consumption of meat [57,62]. Furthermore, one of these studies also reported that a high adherence to a traditional, Western style diet (non-sustainable) was associated with decreased consumption of fruit, vegetables and poultry, and an increased consumption of meat, legumes and wholegrains [57].

The one study that measured change in consumption pre and post intervention reported that at follow up the intervention group had increased intake of fruit, vegetables, legumes, fish and wholegrains and decreased intake of poultry, whereas the control group at follow up had decreased intake of fruit, vegetables, legumes, poultry and fish and increased intake in wholegrains [58]. See Table 5 for further details.

#### 3.4.4. Nutrient Intake

Four out of the 12 studies (n = 4693) reported nutrient intake [59,62,63,64], two studies reported a range of nutrients including energy, carbohydrates, total protein, protein from animal sources, protein from plant sources, and fibre [59,62]. The other two studies reported energy intake only [63,64]. All four studies reported nutrient intake according to level of adherence to sustainable-type diets. Two studies reported that a higher adherence to the diet was associated with a decrease in energy intake [62,63], one study reported very little difference in energy intake between the lowest quartile and highest quartile of dietary adherence [59], and one study reported increases in energy intake for higher adherence to the alternative healthy eating index, the alternative Mediterranean diet and the DASH diet [64].

Two studies (n = 1078) reported that intake of protein from vegetable sources and fibre intake increased, and carbohydrate intake decreased with increasing adherence to diet [59,62]. One study reported increases in total protein and protein from animal sources as adherence to diet increased [59], whereas the other study reported a decrease in total protein and animal protein as adherence to diet increased [62]. It is of note that the latter study measured adherence to a healthy plant based diet [62], whereas the former study measured adherence based on amount of total protein consumed per day [59]. See Table 6 for further details.

#### 3.4.5. Health Status

Nine out of the 12 studies (n = 10,948) reported health measures [57,59,60,62,63,64,66,67,68], with eight studies reporting body mass index (BMI) [57,59,60,62,63,64,66,68], six studies reporting non-communicable diseases such as type 2 diabetes, hypertension, cardiovascular diseases and cancer [57,59,63,64,66,67], three studies reporting self-rated health [57,59,66], two studies reporting cholesterol levels [57,68], and one study reporting muscle mass and grip strength [59].

Of the eight studies that reported BMI , five reported BMI changes according to level of adherence to sustainable-type diets [59,62,63,64,66], with four of these reporting a decrease in mean BMI by 1.3 to 5.5 kg/m^2^ as adherence to a sustainably-sourced diet increased [59,62,64,66]. One of these studies also reported an increase in mean BMI by 1.9 kg/m^2^ as adherence to a Western style diet increased [66]. The other study reporting BMI according to dietary adherence, measured by BMI category (i.e., underweight, healthy BMI, overweight or obese), rather than mean BMI. As adherence to diet increased, authors reported an increase in the number of people with a healthy BMI (<25 kg/m^2^) from 16.6% to 21.4%, a decrease in the number of people with obesity (BMI > 30 kg/m^2^) from 34.0% to 29.9%, and no change in people who were overweight (BMI 25–29.9 kg/m^2^) [63]. The remaining 3 studies that reported on BMI all reported according to different factors. One reported according to levels of 3-OHFAs measured in the participants and reported that as levels of 3-OHFAs increased there was an increase in number of people who were underweight (BMI < 18.5 kg/m^2^) from 0 to 2.2%, a decrease in people with a healthy BMI (18.5–25 kg/m^2^) from 42.2 to 35.2%, and an increase in people with an overweight or obese BMI (>25 kg/m^2^) from 57.8% to 62.6% [57]. One reported no difference between the median BMI of the studies control group (26.8 kg/m^2^) and intervention group (26.9 kg/m^2^) at baseline, prior to the intervention starting [60]. And one study reported no difference between mean BMI of all participants at baseline (24.0 kg/m^2^), 2 months (24.3 kg/m^2^) and 4 months (24.3 kg/m^2^) [68].

Of the six studies that reported non-communicable diseases, five reported according to adherence to sustainable-type diets [59,63,64,66,67]. Three of these five reported on type 2 diabetes with two reporting a decrease in number of people with type 2 diabetes as adherence to diet increases (17.25 to 13.56% [63], 43.0 to 25.0% [64], 38.0 to 25.0% [64]) and one reporting an increase in type 2 diabetes from 11.3 to 16.0% as adherence increases [59]. Two of the five- studies reported on depression but both showed very little change as dietary adherence changed (depression score ranging from 0 to 15: 1.8 to 1.4 [59], and percentage of depression occurrence: 7.69 to 7.87% [63]). One each of the five papers reported on hypertension [59], cardiovascular diseases [63], cancer [63], lung disease [63], history of multimorbidity [66] and death [67] with mixed outcomes. The one remaining study, reported non-communicable diseases according to levels of 3-OHFAs levels measured in the participants and reported that as levels of 3-OHFAs increased, there was a decrease in depression from 6.9 to 5.2% and an increase in type 2 diabetes from 3.5 to 13.3%, metabolic syndrome from 11.4 to 20.7%, hypertension from 75.8 to 80.3%, and cardiovascular diseases from 5.2% to 10.7% [57].

Of the three studies that reported self-rated health, one reported that increases in 3-OHFAs levels resulted in an increase in people reporting poor health from 60.3% to 64.4%, but no change in people reporting excellent health (3.0 to 3.0%) [57]. Another study reported a decrease in people reporting excellent health from 38.7 to 35.1% as total daily protein intake increased [59]. Additionally, one study reported no change in people reporting excellent health from 87.7 to 87.6% as adherence to prudent diet increased [66].

The two studies that reported cholesterol both reported slight increases to low density lipoprotein (mean: 3.6 to 3.7 mmol/L [57], and 102.0 to 103.2 mg/dL [68]) and minimal changes to high density lipoprotein levels (mean: 1.6 to 1.6 mmol/L [57], and 61.2 to 61.8 mg/dL [68]). 

The one study reporting muscle mass and grip strength reported that as daily protein intake increased, muscle mass increased from a mean (SD) of 49.1 (5.6) kg to 76.0 (7.0) kg and hand grip strength decreased from a mean (SD) of 36.4 (7.9) to 35.9 (7.6) kg [59]. See Appendix A for further details.

## 4. Discussion

This review investigated the available evidence for the impact of consumption of sustainably sourced or plant-based diets on gut health, nutrient intake and health status in older community dwelling people. Of the studies that assessed gut health, most found that higher adherence to sustainably sourced diets positively impacted on gut microbiota or the by-products of metabolic processes in bacteria (metabolites), which links with previous work stating that modification to diet can change the bacterial strains dominating the gut so contributing to its dynamic nature [70]. None of the included studies assessed inflammatory markers, although many previous studies carried out with adult populations of all ages have demonstrated a link between certain diets, particularly the Mediterranean diet and high fibre diets, and a decrease in inflammatory markers [20,24,25,71,72]. This review also found a small number of studies demonstrating that the consumption of sustainably sourced diets was associated with a healthier intake of food groups, although the impact on nutrient intake and health status was unclear with studies that reported on these giving mixed results.

Initially the review assessed adherence to sustainably sourced diets, including the Mediterranean diet or indices, the prudent diet, NU-AGE diet (Mediterranean style), amount of daily plant and animal protein intake, healthy plant-based diet index, healthy eating indices, the DASH diet, and plant-based smoothies/snacks. Mostly sustainable dietary compliance was measured from the participants usual dietary intake and in these studies around a quarter of participants (ranging from 19.8 to 36.0%) were found to already be highly compliant with the diet being investigated [57,59,61,62,63,64,65,66,67]. Only three papers out of the 12 assessed used a plant-based dietary intervention and showed much higher compliance of 88.0 to 92.5% in one study [68], although this study was not a whole diet approach intervention, and an increase of 27% compliance [58] and significant increase of fibre, vitamins and minerals [60] in the intervention group of the NU-AGE studies. Despite these results, the lack of studies with sustainable, plant-based dietary interventions in older people, highlights the need for increased investigation in this area.

Although the sustainably sourced diets being investigated across studies varied widely, those that measured microbiota outcomes reported that increased adherence to each of the sustainable-type diets improved gut microbiome diversity or lowered factors such as 3-OHFAs, which are linked to inflammation. This concurs with previous work where the Mediterranean diet has shown promise by beneficially impacting on diversity and richness of gut microbiota [73] and in improving metabolic status of elderly women with obesity [53]. The one study that disagreed, and reported reduced bacterial diversity, was considered to have an overall high risk of bias [68]. This suggests that dietary recommendation or future dietary interventions in older people may not necessarily need to follow a strict pattern or focus solely on increasing protein in order to be effective. Indeed, the intake of other nutrients like fibre and the role they play in maintaining health must also be considered whilst acknowledging over-nutrition can be just as harmful as under nutrition [74]. In addition, a more varied diet in terms of protein source may better serve muscle preservation in older age as different sources of protein and fibre are important for bacterial diversity in the gut microbiome, both of which may underpin protein metabolism, availability and muscle synthesis [14,16,75].

Reporting of nutrient intake and food group intake was relatively low compared to other outcomes with only four studies reporting nutrient intake [59,62,63,64] and three studies reporting food group intake [57,58,62]. The data for food group intake was positive with all three studies showing that higher adherence to sustainable-type diets was associated with higher intake of healthier food groups, including fruits, vegetables, legumes and fish. However, data for nutrient intake was mixed and insufficient to draw any meaningful conclusions. It is worth noting that two studies reported that higher adherence to a healthy plant-based diet or higher daily protein intake was associated with higher intake of fibre and protein from vegetable sources [59,62]. This is important as evidence suggests that insufficient dietary protein and fibre intake can have a negative impact on maintaining muscle strength and physical function as we age [26,27]. 

Results for health status were also very mixed across nine of the included studies, making it difficult to draw meaningful conclusions. In addition, many of the studies reported only on one or two health outcomes and so data was lacking particularly for self-rated health, cholesterol, muscle mass and hand grip strength. However, we know that previous work has demonstrated that both animal and plant proteins can be incorporated into a healthy diet to reduce risk of age-related diseases [76,77] and habitual consumption of polyphenols from fruit, vegetables, nuts, herbs and spices may support an intestinal environment with lower relative abundance of pathogenic bacteria [78]. 

Based on protein quality alone, animal sources would appear to better support muscle mass and maintenance [43], which is particularly important as we age. However, this review further adds to the evidence base and may help us to better understand that encouraging increased consumption of animal proteins may not be the best outcome for overall health as well as the environment [32,39]. It may, for example, be accompanied by a decrease in fibre intake [35] and so reduce diversity of the gut microbiota [7], and an increase in fat intake and consumption of red meat has been linked to increased cardiovascular disease risk [79,80]. The demand for animal protein creates resource intensive farming systems resulting in deforestation, water scarcities, soil depletion and higher greenhouse gas emissions so therefore can have substantial environmental implications [36]. However, moving towards a more sustainable diet that reduces environmental impact without detriment to meeting dietary requirements, does not necessarily have to eliminate meat and dairy products [81] and can also include the option to incorporate alternative, more sustainable protein sources such as fruits, vegetables, insects, mycoproteins, legumes and microbial proteins, [82] which have been shown to have beneficial effects on human health [83]. 

When formulating recommendations regarding protein intake in older age and designing corresponding interventions, it is imperative to account for both the quantity and source of protein, alongside understanding the optimal requirements necessary for older individuals. Accumulating evidence suggests that there is an escalating demand for protein with advancing age, primarily to counteract age related changes in protein metabolism [84]. Moreover, to maintain muscle mass, healthy older adults may require an additional intake of 0.2–0.4 g of protein per kilogram of body weight daily (yielding a total of 1.0–1.2 g per kilogram of body weight) compared to their younger counterparts [85,86]. Furthermore, it is essential to acknowledge potential adverse consequences associated with sustainably sourced diets as indicated by recent systematic review findings suggesting that diets aimed at reducing environmental impact may result in lower intake and status of critical micronutrients across healthy individuals of varying age [87]. Variables that may influence protein bioavailability include the digestibility and postprandial metabolism of different protein sources [88], as well as the composition of the food matrix (e.g., levels of fat, fibre and carbohydrates) [74] and physiological factors that affect metabolism such as health status, physical activity [86], energy equilibrium [74], and sex-related disparities stemming from variations in body composition and hormonal fluctuations (e.g., menopause) [89]. Although there exists scant data concerning the significance of protein source in meeting heightened protein requirements with advancing age, this review underscores the potential utility of sustainably sourced diets in fostering gut health and mitigating inflammation. 

Any future research or dietary interventions aimed at augmenting the consumption of sustainably sourced proteins should, at the forefront, accommodate potential age-specific disparities in requirements and any predisposing nutritional susceptibilities that may be exacerbated by such dietary patterns. It is imperative to recognise the bidirectional relationship between the gut microbiota and inflammation, whereby inflammation and its triggers may emanate from multifaceted interactions within the host, including genetic predispositions, age, sex, mode of birth, antibiotic use, dietary habits, BMI, and infant feeding practices [70]. Consequently, future endeavours should not only contemplate the implementation of sustainably sourced, high protein, and high fibre diets among older populations but also advocate for dietary interventions that are characterised by diversity and a whole diet, holistic approach, supporting sustainable and favourable alterations in overall dietary intake, whilst appropriately considering the aforementioned critical host characteristics.

### Limitations of Evidence and the Review Process

The review was limited mainly by the low number of studies identified that included dietary interventions with most studies measuring dietary adherence from participants usual intake. Without interventions or randomised controlled trials, it is difficult to measure impact and make conclusions. Moreover, as many of the studies investigated different types of sustainable diets and measured different types of outcomes it was not possible to make formal comparison or conduct a meta-analysis. In addition, the lack of a definition for a high protein sustainable diet meant that there were no specific inclusion or exclusion criteria for the diet being investigated and none of the included studies identified the investigated diet as being sustainable. Finally, this review only considered gut microbiota or inflammatory markers as a measure of gut health but a more detailed picture may be gained by taking into account indicators of gut health including early satiety, malabsorption, prolonged transit time, and reduction in both neurotransmitters and receptors.

## 5. Implications for Practice, Policy and Future Research

Maintaining gut health and physical function are an essential part of healthy ageing. There is an increasing need to support the ageing population with intake of sufficient dietary protein and fibre in order to maintain physical function and reduce health issues. This review has highlighted the advantages of supporting sustainable eating in older adults. Future work should consider dietary interventions that use sustainable protein combined with fibre, which can be scaled up to randomised controlled trials. Further investigation may also consider exploring the impact of protein and fibre on gut barrier function, inflammation and body composition, without compromising protein quality and quantity and with an overarching aim of reducing dependence on animal source proteins. Although, as dietary intake will be highly influenced by preference, behaviour, habit and food access, it will be essential to firstly investigate barriers and facilitators to sustainable eating in older people.

## 6. Registration and Protocol

In accordance with the guidelines, our systematic review protocol was registered with the International Prospective Register of Systematic Reviews (PROSPERO) on 5 September 2023 (Registration number: PROSPERO 2023 CRD42023459818 Available from: https://www.crd.york.ac.uk/prospero/display_record.php?ID=CRD42023459818 (accessed on 3 May 2024).

## Figures and Tables

**Figure 1 nutrients-16-01398-f001:**
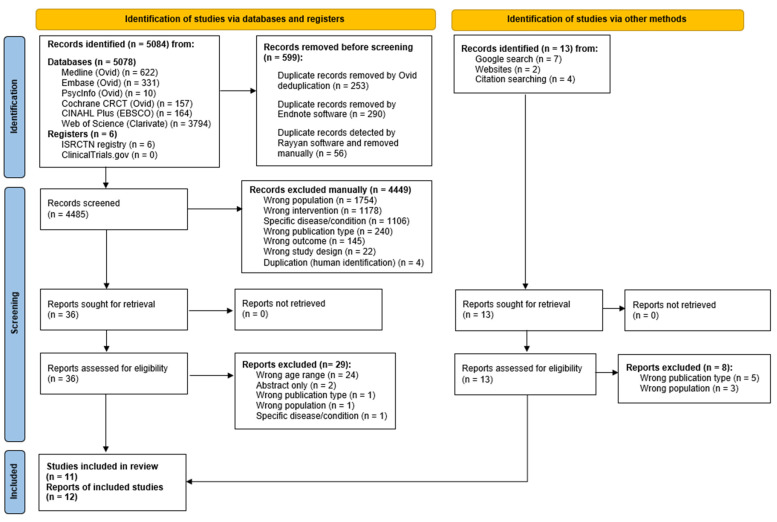
PRISMA flow diagram of study inclusion process [69].

**Table 1 nutrients-16-01398-t001:** Study and participant characteristics of included studies.

Author, Year (Country)	Study Design	Sample Size (n)	Age Mean (SD) Years	Female n (%)	Dietary Intervention/ Exposure	Outcomes	Duration (Months)
André, 2021 [57] (France)	Cross-sectional	698	73.1 (4.4)	432 (61.9)	Mediterranean vs. prudent vs. traditional vs. complex carbs	Circulating 3-OH FAs, a proxy of LPS-type endotoxins burden.	Dietary survey conducted after 24 months
Berendsen, 2018 [58] (The Netherlands) Recruited from five EU centres	Randomised multicentre, single-blind, controlled trial	1141	71.0 (4.0)	631 (55)	Mediterranean-like diet (NU-AGE diet) with counselling and dietary advice vs. control	Dietary intake	12 months follow up
Farsijani, 2022 [59] (USA)	Cross-sectional	775	84.2 (4.0)	0 (0)	Usual food intake to measure total daily protein intake	Microbial DNA extraction from stool sample for gut microbiome profiling (16S rRNA gene sequencing)	Mailed FFQ to MrOS mean of 4.6 (SD 11.7) days after stool collection.
Ghosh, 2020 [60] (Ireland) Recruited from five EU study centres	Randomised multicentre, single-blind, controlled trial	612	71 (range: 65–79)	326 (53)	Mediterranean-like diet (NU-AGE diet) + counselling + dietary advice vs. control	Gut microbiome profile	12 months follow up
Gutierrez-Díaz, 2016 [61] (Spain)	Cross-sectional	74 (50 yrs and above) 50–65 yrs: n = 37 ≥65 yrs: n = 37	71.3 (11.2) (Mean for subgroups not reported)	Not reported for ≥65 years subset. Total sample: 54 (73)	Mediterranean diet score	Anthopometric data, microbiological and phenolic metabolite assessment of fecal specimens.	Cross sectional
Li, 2021 [62] (USA)	Cohort	303	71 (4.0)	0 (0)	hPDI	Gut microbiome profile	12 months follow up
Maroto-Rodriguez, 2022 [63] (Spain)	Cohort	1880	68.65 (6.38)	971 (51.56)	hPDI & uPDI	Health outcomes recorded at baseline and frailty status recorded at follow up	Follow up was on average 40 months
Maskarinec, 2019 [64] (USA)	Cohort	1735	69.2 (at stool collection)	877 (50.5)	HEI-2010, AHEI-2010, aMED, and DASH diet	Association of diet quality with measures of stool microbial community structure.	Main analysis is cross sectional
Ruiz-Saavedra, 2020 [65] (Spain)	Cross-sectional	40 (≥65 yrs) Total sample: n = 73	2 sub-groups: 50–65 years and 65–95 years (mean not reported)	Not reported for ≥65 subset. Total sample: 53 (73)	DII, EDII, HEI, AHEI, DQI-I, MMDS, and rMED	Major phylogenetic types of the intestinal microbiota determined by qPCR and SCFAs	Cross-sectional
Shikany, 2019 [66] (USA)	Cross-sectional	517	84.3 (4.1)	0 (0)	2 dietary patterns: Western and Prudent.	Diversity of gut bacterial microbiota	Dietary assessment completed within a mean (SD) 4.6 (11.7) days of the stool sample collection
Trichopoulou, 2003 [67] (Greece)	Cohort	4369 (≥65 yrs) Total sample n = 22,043	3 sub-groups: <55 years, 55–64 years, and ≥65 years (mean not reported)	Not reported for ≥65 subset. Total 13,143 (60)	Mediterranean diet	Mortality	44 months follow up
Zhang, 2021 [68] (Taiwan)	Cohort	59	Female: 77.7 (7.6) Male: 85.3 (8.4)	30 (51)	Plant-based, antioxidant-rich smoothies and sesame seed snacks	Antioxidant ability and gut microbial composition	Follow up at 2 and 4 months

n: number, SD: standard deviation, 3-OHFAs: 3-hydroxy fatty acids, hPDI: healthy plant-based diet index, uPDI: unhealthy plant-based diet index, HEI-2010: Healthy Eating Index 2010, AHEI-2010: Alternative Healthy Eating Index 2010, aMED: Alternate Mediterranean Diet, DASH diet: Dietary Approaches to Stop Hypertension Trial diet, DII: Dietary inflammatory index, EDII: Empirical Dietary Inflammatory Index, HEI: Healthy Eating Index, AHEI: Alternative Healthy Eating Index, DQI-I: Mediterranean adapted Diet Quality Index-International, MMDS: Modified Mediterranean Diet Score, rMED: relative Mediterranean Diet Score, qPCR: real time polymerase chain reaction, SCFAs: short chain fatty acids.

**Table 2 nutrients-16-01398-t002:** Intervention characteristics and adherence to diet.

Author, Year	Dietary Measurement Tool	Dietary Pattern Assessment	Adherence to Diet	Main Findings
André, 2021 [57]	FFQ (By registered dietitian)	Med diet: Score ranged from 0, low adherence to 18, high adherence. Carbs/traditional/prudent diet: Factor analysis with tertile range (upper tertile = higher adherence).	Med diet: Mean score: 10.7 (SD 2.0) n = 698 Low (<9): n = 187 (27.0%) Medium (10–12): n = 264 (38.0%) High (>12): n = 247 (35.0%) Carbs/traditional/prudent diet: Low: n = 232 (33.2%) Medium: n = 233 (33.4%) High: n = 233 (33.4%)	Plant-based dietary patterns were associated with lower 3-OH FA concentrations, and thus a lower LPS burden, which is considered a potent trigger of inflammatory response.
Berendsen, 2018 [58]	Self-reported 7-day records (with prior training)	NU-AGE index score, with diet compliance ranging from 0 (low) to 160 (high).	Baseline mean score (SD): Control group: 82.6 (16.5) Diet group: 82.6 (15.3) Follow-up mean score (SD): Control group: 84.6 (16.1) Diet group: 105.7 (17.6)	The NU-AGE dietary intervention may be a feasible strategy to improve dietary intake in an aging European population.
Farsijani, 2022 [59]	Self-reported Brief FFQ (Block 98.2 MrOS)	Total daily protein intake (g/d) was estimated from the collected FFQs. Data was recorded by quartile of energy adjusted protein intake.	Q1: ≤55.44 g/d, n = 194 (25.0%) Q2: 55.45–61.17 g/d, n = 193 (24.9%) Q3: 61.18–67.98 g/d, n = 194 (25.0%) Q4: ≥67.99 g/d, n = 194 (25.1%)	Higher protein consumptions from either animal or vegetable sources were associated with higher gut microbiome diversity.
Ghosh, 2020 [60]	Self-reported 7-day records (with prior training)	NU-AGE index score, with diet compliance ranging from 0 (low) to 160 (high).	Dietary variations within the intervention group were significantly different from the control group (envfit *p* < 0.006). Intervention group: increased intake of fibres, vitamins (C, B6, B9, thiamine) and minerals (Cu, K, Fe, Mn, Mg). Control: Increase in fat intake (saturated fats and mono-unsaturated fatty acids) relative to the intervention group.	Increasing adherence to the NU-AGE diet was associated with higher gut microbiome diversity
Gutierrez-Díaz, 2016 [61]	FFQ 24 h dietary intake	Med Diet Score (MDS) calculated based on eight dietary components, with a possible range of 0–8 points. Cut-off for greater adherence and health-promoting effects was 4 points.	Total sample (50 years and above) MDS < 4 points: n = 32 (43%) MDS ≥ 4 points: n = 42 (57%) Diet scores not reported for >65 years subgroup	Older subjects (>65 years), and subjects with sedentary habits exhibited higher values for the fecal content of phenylacetic, 4-hydroxyphenylacetic, and phthalic acids.
Li, 2021 [62]	Validated FFQ	hPDI was derived from FFQ. Scores were summed to give a hPDI score of 18 (lowest) to 90 (highest).	hPDI score, mean (SD): Q1: 46.5 (2.6) n = 59 (19.6%) Q2: 51.3 (0.9) n = 62 (20.4%) Q3: 54.4 (0.9) n = 60 (19.8%) Q4: 58.0 (1.2) n = 62 (20.4%) Q5: 64.1 (2.8) n = 60 (19.8%)	A greater adherence to a healthy plant-based diet was associated with a microbial profile characterized by a higher abundance of multiple species.
Maroto-Rodriguez, 2022 [63]	A validated computerised face-to-face diet history (DH-ENRICA) developed from EPIC cohort study in Spain	hPDI was derived from FFQ. Scores were summed to give hPDI and uPDI scores of 18 (lowest) to 90 (highest) and then categorized into 3 tertiles.	hPDI, mean score (SD): Total: 59.73 (5.73) n = 1880 T1: 52.43 (2.62) n = 429 (23%) T2: 58.60 (1.71) n = 765 (41%) T3: 65.56 (3.22) n = 686 (36%) uPDI, mean score (SD): Total: 54.85 (5.32) n = 1880 T1: 50.32 (3.13) n= 879 (47%) T2: 56.83 (1.37) n = 639 (34%) T3: 62.38 (2.52) n = 362 (19%)	In older adults, the hPDI was associated with lower risk of frailty, while the opposite was found for the uPDI.
Maskarinec, 2019 [64]	QFFQ covering over 180 food items	Scores for 4 diets were calculated. Score ranges shown in next column. In all 4 diets the higher the score the higher the adherence to the diet.	HEI-2010, score range: T1: 35.2—68.4, n = 578 (33.3%) T2: 68.5—77.7, n = 579 (33.4%) T3: 77.8—99.1, n = 578 (33.3%) AHEI-2010, score range: T1: 35.6—64.5, n = 578 (33.3%) T2: 64.6—73.0, n = 579 (33.4%) T3: 73.1—99.4, n = 578 (33.3%) aMED, score range: T1: 0–3, n = 643 (37.1%) T2: 4–5, n = 627 (36.1%) T3: 6–9, n = 465 (26.8%) DASH, score range: T1: 9–22, n = 631 (36.4%) T2: 23–26, n = 545 (31.4%) T3: 27–38, n = 559 (32.2%)	Diet quality was strongly associated with fecal microbial alpha diversity and beta diversity and several genera previously associated with human health
Ruiz-Saavedra, 2020 [65]	Semi-QFFQ	Scores for 7 dietary indices were calculated for the total sample (aged 50–95 years).	Subgroup >65 years, mean score (SD) (n = 40): DII: 0.98 (2.02) EDII: 1.02 (0.69) HEI: 54.46 (10.16) AHEI: 58.39 (6.88) DQI-I: 46.60 (5.96) MMDS: 3.13 (1.49) rMED: 6.15 (2.03)	DII, HEI, DQI-I and MMDS were identified as predictors of Faecalibacterium prausnitzii levels, AHEI and MMDS were negatively associated with Lactobacillus group. HEI, AHEI and MMDS were positively associated with fecal SCFAs.
Shikany, 2019 [66]	Block 98.2 MrOS FFQ (NutritionQuest)	Final factor scores were calculated through analysis of the FFQs. Adherence to the dietary patterns was divided into quartiles, with Quartile 1 representing the lowest adherence and Quartile 4 representing the highest adherence.	Western diet: Q1: n = 130 (25.0%) Q2: n = 129 (25.0%) Q3: n = 129 (25.0%) Q4: n = 129 (25.0%) Prudent diet Q1: n = 130 (25.0%) Q2: n = 129 (25.0%) Q3: n = 129 (25.0%) Q4: n = 129 (25.0%)	Significant associations between measures of gut microbial composition and dietary patterns.
Trichopoulou, 2003 [67]	Semi-QFFQ administered by specially trained interviewers	Mediterranean-diet score ranged from 0 (minimal adherence to the traditional Mediterranean diet) to 9 (maximal adherence)	Subgroup >65 years, Med diet score range: T1: Score 0–3, n = 1598 (36.6%) T2: Score 4–5, n = 1829 (41.9%) T3: Score 6–9, n = 942 (21.5%)	Greater adherence to the traditional Mediterranean diet is associated with a significant reduction in total mortality
Zhang, 2021 [68]	Consumption of specified smoothies and snacks	Each serving of a plant-based smoothie contained 1 exchange of vegetables (2 kinds), 1 exchange of fruits (2 kinds), and 1 exchange of nuts.	All participants were provided with 5 servings of plant-based smoothies and 3 servings of sesame seed snacks per week. Participants received these for a 4-month period.	Consumption of Plant-based smoothies and snacks prompted significant decreases in observed bacterial species and their richness.

FFQ: food frequency questionnaire, Med: Mediterranean, Carbs: Carbohydrates, Q: quartile/quintile, hPDI: healthy plant-based diet index, uPDI: unhealthy plant-based diet index, T: tertile, QFFQ: quantitative food frequency questionnaire, HEI-2010: Healthy Eating Index 2010, AHEI-2010: Alternative Healthy Eating Index 2010, aMED: Alternate Mediterranean Diet, DASH diet: Dietary Approaches to Stop Hypertension Trial diet, DII: Dietary inflammatory index, EDII: Empirical Dietary Inflammatory Index, HEI: Healthy Eating Index, AHEI: Alternative Healthy Eating Index, DQI-I: Mediterranean adapted Diet Quality Index-International, MMDS: Modified Mediterranean Diet Score, rMED: relative Mediterranean Diet Score, MrOS: Osteoporotic Fractures in Men.

**Table 3 nutrients-16-01398-t003:** Summary of the risk of bias in studies.

RoB 2.0	D1	D2	D3	D4	D5	Overall		
Berendsen, 2018 [58]	Low	Low	Low	Low	Low	Low		
Ghosh, 2020 [60]	Low	Low	Low	Low	Low	Low		
**ROBINS-E**	**D1**	**D2**	**D3**	**D4**	**D5**	**D6**	**D7**	**Overall**
Andre, 2021 [57]	Low	Some concerns	Low	Low	Low	Low	Low	Some concerns
Farsijani, 2022 [59]	Some concerns	Some concerns	Low	Low	Low	Low	Low	Some concerns
Gutierrez-Diaz, 2016 [61]	Some concerns	Some concerns	Low	Low	Low	Low	Low	Some concerns
Li, 2021 [62]	Some concerns	Some concerns	Low	Low	Low	Low	Low	Some concerns
Maroto-Rodriguez, 2022 [63]	Low	Some concerns	Low	Low	Low	Low	Low	Some concerns
Maskarinec, 2019 [64]	Low	Some concerns	Low	Low	Low	Low	Low	Some concerns
Ruiz-Saavedra, 2020 [65]	Low	Some concerns	Low	Low	Low	Low	Low	Some concerns
Shikany, 2019 [66]	Some concerns	Some concerns	Low	Low	Low	Low	Low	Some concerns
Trichopoulou, 2003 [67]	Low	Some concerns	Low	Low	Low	Low	Low	Some concerns
Zhang, 2021 [68]	Some concerns	High	Low	Low	Low	Low	Low	High

Rob 2.0; D1: Bias arising from randomization process, D2: Bias due to deviations from intended intervention, D3: Bias due to missing outcome data, D4: Bias in measurement of the outcomes, D5: Bias in selection of the reported result.; ROBINS-E; D1: Bias due to confounding, D2: Bias arising from measurement of the exposure, D3: Bias in selection of participants into study, D4: Bias due to post-exposure interventions, D5: Bias due to missing data, D6: Bias arising from measurement of the outcome, D7: Bias in selection of the reported result.

**Table 4 nutrients-16-01398-t004:** Microbiota outcomes.

Author, Year	Microbiology
André, 2021 [57]	3-OHFAs pmol/mL (SD) Med diet adherence: Low: 276.7 (110.4), Medium: 261.8 (92.9), High: 263.7 (99.7) Comp carbs adherence: Low: 270.6 (118.7), Medium: 266.1 (87.2), High: 262.6 (92.6) Trad diet adherence: Low: 249.9 (83.4), Medium: 258.6 (91.4), High: 290.8 (118.4) Prudent diet adherence: Low: 283.0 (114.6), Medium: 261.1 (85.2), High: 255.3 (97.3)
Farsijani, 2022 [59]	α-diversity Higher protein intake from vegetable sources compared to lower intake from vegetable sources was associated with higher Chao1 and Shannon indices (overall *p* < 0.05). Higher protein intake from animal sources compared to lower intake from animal sources was associated with higher Shannon and Inverse Simpson indices (overall *p* < 0.05).
Ghosh, 2020 [60]	Taxonomies Adherence to the Mediterranean diet led to increased abundance of specific taxa that were positively associated with several markers of lower frailty and improved cognitive function, and negatively associated with inflammatory markers including C-reactive protein and interleukin-17.
Gutierrez-Díaz, 2016 [61]	Phenolic profiles (μg/mL) Subgroup age ≥ 65 yrs (n = 37): Phenylacetic acid: 16.56 (20.38) Phenylpropionic acid: 10.03 (9.82) Benzoic acid: 0.54 (0.97) 3-hydroxyphenylacetic acid: 0.22 (0.28)
Li, 2021 [62]	Taxonomies A higher hPDI score was significantly associated with 7 microbial species, with: Higher relative abundance (%) of: *Bacteroides cellulosilyticus* (2.58%; 95% CI: 1.39, 3.77) and *Eubacterium eligens* (1.37%; 95% CI: 0.55, 2.20) Lower abundance (%) of: *Ruminococcus torques* (−1.09%; 95% CI: −1.67, −0.50), *Ruminococcus gnavus* (−1.10%; 95% CI: −1.69, −0.52), *Clostridium leptum* (−0.66%; 95% CI: −1.03, −0.30), *Lachnospiraceae bacterium* 1_4_56faa (−0.29%; 95% CI: −0.45, −0.12), and *Erysipelotrichaceae bacterium* 21_3 (−0.12%; 95% CI: −0.18, −0.05)
Maskarinec, 2019 [64]	α-diversity (Shannon), mean (95% CI) HEI-2010, T1: 6.03 (5.89, 6.17), T2: 6.15 (6.01, 6.28), T3: 6.15 (6.01, 6.29) AHEI-2010, T1: 6.02 (5.88, 6.15), T2: 6.13 (6.00, 6.27), T3: 6.14 (6.00, 6.28) aMED, T1: 6.05 (5.91, 6.18), T2: 6.11 (5.97, 6.24), T3: 6.16 (6.02, 6.31) DASH, T1: 6.07 (5.94, 6.21), T2: 6.11 (5.98, 6.25), T3: 6.17 (6.03, 6.31) Phylum Actinobacterium, mean (95% CI) HEI-2010, T1: 2.01 (1.80, 2.23), T2: 1.69 (1.47, 1.90), T3: 1.65 (1.43, 1.87) AHEI-2010, T1: 1.91 (1.69, 2.12), T2: 1.78 (1.56, 1.99), T3: 1.67 (1.45, 1.88) aMED, T1: 1.98 (1.76, 2.19), T2: 1.70 (1.49, 1.91), T3: 1.60 (1.37, 1.82) DASH, T1: 1.94, (1.72, 2.16), T2: 1.73 (1.52, 1.95), T3: 1.63 (1.41, 1.86)
Ruiz-Saavedra, 2020 [65]	Microbiological parameters: Significant differences were observed in most of the microbiological parameters analyzed according to age. Subjects over 65 years of age presented lower fecal levels of Bacteroides-Prevotella-Porphyromonas group, Clostridia cluster XIVa and Faecalibacterium, as well as all the short chain fatty acids determined. Blood parameters are within the normal physiological ranges and were similar between the groups evaluated except for MDA, IL-8, IL-12 and TNF-α, whose concentration is higher in subjects over 65 years of age. Microbial levels for sub group >65 years: Bacteroides-Prevotella-Porphyromonas (Log10 n ◦ cells/g feces): 8.79 (0.69) Clostridia cluster XIVa (Log10 n ◦ cells/g feces): 6.45 (1.54) *Faecalibacterium prausnitzii* (Log10 n ◦ cells/g feces): 6.42 (1.31) Acetic acid (mM): 23.18 (14.45) Propionic acid (mM): 9.50 (7.46) Butyric acid (mM): 8.44 (7.94)
Shikany, 2019 [66]	α-diversity, mean (SD) Western diet: Shannon: Total sample: 3.39 (0.61), Q1: 3.42 (0.66), Q2: 3.43 (0.62), Q3: 3.38 (0.60), Q4: 3.32 (0.57) Inverse Simpson: Total sample: 15.9 (9.8), Q1: 16.9 (10.2), Q2: 16.6 (10.8), Q3: 15.9 (9.8), Q4: 14.3 (8.3) Prudent diet: Shannon: Total sample: 3.39 (0.61), Q1: 3.38 (0.63), Q2: 3.44 (0.59), Q3: 3.36 (0.61), Q4: 3.38 (0.62) Inverse Simpson: Total sample: 15.9 (9.8), Q1: 15.9 (9.8), Q2: 16.8 (10.3), Q3: 15.4 (10.0), Q4: 15.6 (9.4) Beta-diversity In multivariable-adjusted models, greater adherence to the Western pattern was positively associated with families Mogibacteriaceae and Veillonellaceae and genera Alistipes, Anaerotruncus, CC115, Collinsella, Coprobacillus, Desulfovibrio, Dorea, Eubacterium, and Ruminococcus, while greater adherence to the prudent pattern was positively associated with order Streptophyta, family Victivallaceae, and genera Cetobacterium, Clostridium, Faecalibacterium, Lachnospira, Paraprevotella, and Veillonella. Beta diversity measures were significantly associated with both Western and prudent patterns in multivariable-adjusted analyses.
Zhang, 2021 [68]	α-diversity and SCFAs, at baseline, month 2 and month 4, mean (SD) Acetic acid (µmol/g): 40.98 (16.83), 38.59 (17.86), 30.10 (17.26) Propionic acid (µmol/g) 40.88 (19.21), 42.26 (20.48), 38.89 (20.85) Butyric acid (µmol/g) 36.06 (18.20), 31.61 (16.76), 34.21 (19.06) Chao1: 391.1 (112.5), 301.2 (85.4), 310.2 (77.9) ACE: 387.9 (111.4), 294.6 (78.9), 308.8 (77.9) Shannon 5.09 (0.74), 4.98 (0.68), 5.00 (0.66) Simpson 0.92 (0.06), 0.92 (0.05), 0.93 (0.04) Changes in SCFA content in the feces were not significantly different after 2 and 4 months of intervention Gut microbiota, at baseline, month 2 and month 4, mean (SD) After adjusting for age, gender, and intervention compliance, the older adults were found to have significantly decreased levels of the following bacterial taxa: class Bacilli, genus Streptococcus, genus Ruminiclostridium_5, class Deltaproteobacteria, phylum Actinobacteria, class Bifidobacteriales, and phylum Patescibacteria and increased levels of genus Lactobacillus after 2 and 4 months relative to the baseline. There were significant increases in the phylum Bacteroidetes and species Bacteroides thetaiotaomicron after 2 months and in the genus Agathobacter after 4 months relative to the baseline. There were no appreciable differences in the ratios of Firmicutes and Bacteroidetes—an indicator that is strongly associated with several diseases—at the baseline and 2 and 4 months after the commencement of the intervention (6.6, 4.0, and 6.85, respectively).

3-OHFAs: 3-hydroxy fatty acids, Med: Mediterranean, Comp carbs: Complex Carbohydrates diet, Trad: Traditional, HEI-2010: Healthy Eating Index 2010, AHEI-2010: Alternative Healthy Eating Index 2010, aMED: Alternate Mediterranean Diet, DASH diet: Dietary Approaches to Stop Hypertension Trial diet, T: tertile.

**Table 5 nutrients-16-01398-t005:** Food group outcomes.

Author, Year	Measure of Food Group	Fruit, Mean (SD)	Veg, Mean (SD)	Legumes, Mean (SD)	Poultry, Mean (SD)	Meat, Mean (SD)	Fish, Mean (SD)	Wholegrains/Bread/Cereal, Mean (SD)
André, 2021 [57]	adherence to Med diet, servings/ week adherence to Comp carbs diet, servings/ week adherence to Trad diet, servings/ week adherence to Prudent diet, servings/week	Low: 9.1 (6.6) Medium: 13.4 (6.6) High: 15.6 (5.4) Low: 12.9 (7.3) Medium: 13.3 (6.8) High: 13.0 (6.1) Low: 14.6 (7.1) Medium: 13.2 (6.2) High: 11.5 (6.5) Low: 10.2 (6.1) Medium: 12.6 (5.9) High: 16.3 (6.8)	Low: 8.4 (4.0) Medium: 9.9 (4.1) High: 11.8 (4.0) Low: 9.4 (4.0) Medium: 10.0 (4.0) High: 11.1 (4.5) Low: 10.2 (4.3) Medium: 10.5 (4.2) High: 9.8 (4.1) Low: 7.3 (3.5) Medium: 9.8 (2.9) High: 13.4 (3.6)	Low: 0.5 (0.5) Medium: 0.6 (0.7) High: 0.7 (0.6) Low: 0.4 (0.4) Medium: 0.6 (0.6) High: 0.8 (0.7) Low: 0.3 (0.4) Medium: 0.6 (0.5) High: 0.9 (0.8) Low: 0.6 (0.7) Medium: 0.6 (0.5) High: 0.6 (0.6)	Low: 1.6 (1.1) Medium: 1.8 (1.4) High: 1.9 (1.1) Low: 1.3 (0.9) Medium: 1.7 (1.0) High: 2.5 (1.5) Low: 1.9 (1.4) Medium: 1.8 (1.2) High: 1.7 (1.1) Low: 1.6 (1.1) Medium: 1.8 (1.2) High: 2.1 (1.4)	Low: 5.8 (3.0) Medium: 4.4 (2.4) High: 4.4 (1.9) Low: 4.8 (2.8) Medium: 5.0 (2.3) High: 4.5 (2.3) Low: 3.4 (2.0) Medium: 4.6 (1.9) High: 6.3 (2.6) Low: 5.0 (2.7) Medium: 5.0 (2.5) High: 4.3 (2.3)	Low: 1.9 (1.3) Medium: 2.9 (1.7) High: 3.6 (1.6) Low: 2.2 (1.4) Medium: 3.0 (1.5) High: 3.5 (1.8) Low: 2.8 (1.8) Medium: 2.9 (1.6) High: 2.9 (1.6) Low: 2.3 (1.3) Medium: 2.9 (1.7) High: 3.5 (1.7)	Low: 17.1 (6.0) Medium: 18.6 (5.0) High: 19.4 (4.3) Low: 16.5 (6.0) Medium: 19.1 (4.6) High: 19.9 (4.1) Low: 15.7 (6.2) Medium: 19.4 (4.2) High: 20.4 (3.3) Low: 16.7 (5.6) Medium: 19.0 (4.6) High: 19.8 (4.7)
Berendsen, 2018 [58]	Control group, g/day Diet group (Med style diet), g/day	Baseline: 260.0 (158.7) Follow up: 255.7 (154.0) Baseline: 248.2 (140.2) Follow up: 268.2 (140.0)	Baseline: 221.4 (120.7) Follow up: 213.2 (125.7) Baseline: 214.5 (110.8) Follow up: 234.2 (103.7)	Baseline: 11.1 (20.0) Follow up: 10.8 (19.2) Baseline: 10.4 (20.9) Follow up: 17.6 (21.9)	Baseline: 41.2 (33.4) Follow up: 40.5 (31.4) Baseline: 40.5 (31.6) Follow up: 38.5 (27.9)	N/A	Baseline: 28.4 (29.3) Follow up: 24.9 (23.2) Baseline: 28.4 (25.3) Follow up: 37.1 (28.1)	Baseline: 54.4 (53.9) Follow up: 62.6 (60.7) Baseline: 55.7 (58.3) Follow up: 107.2 (66.4)
Li, 2021 [62]	adherence to hPDI, servings/ day	Q1: 1.3 (0.8) Q2: 1.6 (0.6) Q3: 1.7 (0.8) Q4: 1.9 (1.1) Q5: 2.6 (1.4)	Q1: 3.2 (1.0) Q2: 3.4 (1.4) Q3: 3.5 (1.7) Q4: 3.9 (1.7) Q5: 4.6 (1.6)	Q1: 0.4 (0.2) Q2: 0.5 (0.2) Q3: 0.4 (0.2) Q4: 0.4 (0.2) Q5: 0.7 (0.5)	Not reported	Q1: 1.6 (0.5) Q2: 1.4 (0.5) Q3: 1.3 (0.6) Q4: 1.1 (0.4) Q5: 0.7 (0.4)	Q1: 0.3 (0.1) Q2: 0.4 (0.2) Q3: 0.3 (0.2) Q4: 0.4 (0.2) Q5:0.4 (0.2)	Q1: 1.6 (0.7) Q2: 1.7 (0.9) Q3: 2.1 (1.2) Q4: 1.9 (0.9) Q5:2.5 (1.7)

Med: Mediterranean, Comp carbs: Complex Carbohydrates diet, Trad: Traditional, hPDI: healthy plant-based diet index, Q: quintile, N/A: not applicable.

**Table 6 nutrients-16-01398-t006:** Nutrient outcomes.

Author, Year	Measure of Nutrient Group	Energy, Mean (SD) (kcal/d)	Fat, Mean (SD) (% of kcal/d)	Carbohydrate, Mean (SD) (g/d)	Total Protein, Mean (SD) (g/d)	Veg/Plant Protein, Mean (SD) (g/d)	Animal Protein, Mean (SD) (g/d)	Fibre, Mean (SD) (g/d)
Farsijani, 2022 [59]	Total daily protein intake by quartile	Q1: 1710 (695) Q2: 1372 (568) Q3: 1355 (482) Q4: 1700 (622) Total sample: 1534 (620)	Q1: 41.8 (7.1) Q2: 40.2 (6.8) Q3: 41.3 (7.2) Q4: 40.0 (6.9)	% of kcal/d: Q1: 48.1 (7.5) Q2: 47.4 (7.0) Q3: 44.3 (7.4) Q4: 43.0 (6.9)	Q1: 55.3 (23.5) Q2: 52.3 (21.2) Q3: 57.4 (17.9) Q4: 81.8 (26.1) Total sample, 62.0 (10.8)	Q1: 25.7 (12.5) Q2: 22.1 (10.3) Q3: 21.8 (10.5) Q4: 28.8 (13.4)	Q1: 29.6 (15.1) Q2: 30.2 (13.8) Q3: 35.6 (11.8) Q4: 53.0 (18.4)	Q1: 16.1 (7.9) Q2: 14.9 (6.7) Q3: 14.7 (7.1) Q4: 18.8 (9.0)
Li, 2021 [62]	Adherence to a high protein diet by quintile	Q1: 2353 (411) Q2: 2254 (512) Q3: 2069 (481) Q4: 1987 (510) Q5: 1921 (459)	Not reported	Q1: 275.0 (56.8) Q2: 273.0 (70.7) Q3: 251.0 (64.5) Q4: 241.0 (69.2) Q5: 254.0 (76.7)	Q1: 96.0 (17.5) Q2: 92.7 (23.9) Q3: 90.2 (23.7) Q4: 85.5 (20.8) Q5: 81.0 (18.7)	Q1: 28.6 (6.1) Q2: 30.3 (8.6) Q3: 30.1 (8.9) Q4: 30.0 (8.9) Q5: 35.0 (12.7)	Q1: 67.5 (14.5) Q2: 62.3 (17.4) Q3: 60.0 (17.3) Q4: 55.5 (14.1) Q5: 46.0 (16.2)	Q1: 21.8 (5.5) Q2: 24.5 (7.4) Q3: 24.1 (7.4) Q4: 25.2 (7.9) Q5: 30.4 (10.6)
Maroto-Rodriguez, 2022 [63]	Adherence to a high protein diet by tertile	T1: 2335 (550) T2: 2053 (554) T3: 1815 (499) Total sample, 2031 (569)	Not reported	Not reported	Not reported	Not reported	Not reported	Not reported
Maskarinec, 2019 [64]	Adherence to HEI, AHEI, aMED and DASH by tertile	HEI-2010: T1: 1957 (1050) T3: 1779 (766) AHEI-2010: T1: 1760 (887) T3: 1967 (833) aMED: T1: 1463 (615) T3: 2450 (1136) DASH: T1: 1618 (699) T3: 2109 (1082)	Not reported	Not reported	Not reported	Not reported	Not reported	Not reported

Kcal/d: kilocalories per day, SD: standard deviation, %: percentage, g/d: grams per day, Q: quartile/quintile, T: tertile, HEI-2010: Healthy Eating Index 2010, AHEI-2010: Alternative Healthy Eating Index 2010, aMED: Alternate Mediterranean Diet, DASH diet: Dietary Approaches to Stop Hypertension Trial diet.

## Data Availability

The original contributions presented in this review are included in the article/Appendix A. The initial data extraction and template data extraction forms will be made available upon request and any further inquiries can be directed to the corresponding author.

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
