# Peer review of "Effect of Sustainably Sourced Protein Consumption on Nutrient Intake and Gut Health in Older Adults: A Systematic Review"

_nutrients, 2024, doi:10.3390/nu16091398_

Round 1

Reviewer 1 Report

Comments and Suggestions for Authors

This systematic review of universal healthy sustainable diets provides a PRISMA flow diagram for study inclusion.

The search strategy statistical methodology, and data summaries (Tables 1-6) are appropriate.

The discussion is appropriate including evidence presented, limitations, and gaps in data.

The conclusion is appropriate suggesting dietary interventions related to sustainable eating.

Line 640: should be "counteract"

Author Response

Thank you very much for taking the time to review our manuscript and for your positive comments.

Line 640 has been amended from 'counter-act' to 'counteract'. Thank you.

Reviewer 2 Report

Comments and Suggestions for Authors

This review aims to assess the impact of diets high in sustainably sourced proteins on nutrient intake, gut, and age-related health in older adults. Studies assessing sustainably sourced protein consumption in community dwelling older adults (≥ 65 years) were included. Twelve studies involving 12,166 older adults were included. Increased adherence to sustainably sourced diets was associated with improved gut microbial factors (n=4640), healthier food group intake (n=2142), and increased fibre and vegetable protein intake (n=1078). Sustainably sourced diets positively impacted on gut microbiota and healthier intake of food groups, although effects on inflammatory outcomes and health status were inconclusive. 

I personally dislike systematic reviews because there are so many out there and many of them are similar. However, this systematic review is very readable and summarizes the objectives, methods, and data of each study very clearly. It also descriptively discusses the limitations of this study.

1. This review was primarily concerned with the small number of studies identified that included dietary interventions, and most studies measured dietary adherence from participants' usual intake. Without interventions or randomized controlled trials, it is difficult to measure impact and draw conclusions.

2. Because many studies investigated different types of sustainable diets and measured different types of outcomes, no formal comparisons or meta-analysis could be made.

3. Because there is no definition of a high-protein sustainable diet, there were no specific inclusion or exclusion criteria for the diets studied, and no studies identified the included diets as sustainable.

4. This review considered only gut microbiota or inflammatory markers as indicators of gut health, but a more detailed picture could be obtained by considering other indicators of gut health, such as early satiety, malabsorption, prolonged transit time, and decreases in both neurotransmitters and receptors.

Even with these limitations, I believe the importance of this paper is clear in my opinion.

Author Response

Thank you very much for taking the time to review our manuscript. We are grateful for your positive comments and for stating that you feel the paper is of importance.